# Features of Metabolic Support of Physical Performance in Highly Trained Cross-Country Skiers of Different Qualifications during Physical Activity at Maximum Load

**DOI:** 10.3390/cells11010039

**Published:** 2021-12-23

**Authors:** Olga I. Parshukova, Nina G. Varlamova, Natalya N. Potolitsyna, Aleksandra Y. Lyudinina, Evgeny R. Bojko

**Affiliations:** Institute of Physiology of Komi Science Centre of the Ural Branch of the Russian Academy of Sciences, FRC Komi SC UB RAS, 50 Pervomayskaya Str., 167982 Syktyvkar, Russia; nivarlam@physiol.komisc.ru (N.G.V.); potol_nata@list.ru (N.N.P.); salu_06@inbox.ru (A.Y.L.); boiko60@inbox.ru (E.R.B.)

**Keywords:** nitric oxide, lactate, heart rate, oxygen uptake, arterial blood pressure, exercise test on a cycle ergometer, cross-country skier

## Abstract

The purpose of our study was to identify the features of metabolic regulation in highly trained cross-country skiers of different qualifications at different stages of the maximum load test. We examined 124 highly trained cross-country skiers (male, ages 17–24). The group consisted of two subgroups based on their competition performance: 61 nonelite athletes (Group I) and 63 elite athletes (group II), who were current members of the national team of the Komi Republic and Russia. The bicycle ergometer test was performed by using the OxyconPro system (Erich Jaeger, Hoechberg, Germany). All the examined athletes performed the exercise test on a cycle ergometer “until exhaustion”. The results of our research indicate that the studied groups of athletes with high, but different levels of sports qualifications are a convenient model for studying the molecular mechanisms of adaptation to physical loads of maximum intensity. Athletes of higher qualifications reveal additional adaptive mechanisms of metabolic regulation, which is manifested in the independence of serum lactate indicators under conditions of submaximal and maximum power from maximal oxygen uptake, and they have an NO-dependent mechanism for regulating lactate levels during aerobic exercise, including work at the anaerobic threshold.

## 1. Introduction 

Moderate physical activity has a positive effect on the morphology and work of the cardiovascular system of athletes due to the manifested adaptive response of the myocardium [1]. However, at the same time, a violation in the consistency of the functional state of the system associated with the work of the heart rhythm leads to an overstrain of the cardiovascular system of athletes [2]. Systematically exercising athletes usually develop myocardial hypertrophy. Pathological hypertrophy is based on dystrophic changes in the myocardium and deterioration of the microvasculature, which leads to difficulty in contraction of the left ventricular wall, which ultimately affects the decrease in the athletic performance of the body [3]. Cross-country skiers have a very high maximal oxygen uptake, and they are able to perform submaximal exercise at a rather high metabolic rate, and with cardiac output levels similar to or higher than the cardiac output levels achieved by untrained humans at maximal exercise [4]. The role of the molecular gas nitric oxide (NO) in the cardiovascular system is well established, where NO regulates a multitude of cellular processes. Endothelial cells synthesize and release NO, which mediates diverse effects, including vessel tone, haemostasis, blood pressure and vasculature remodelling [5]. The significance of NO for cardiomyocyte function is well known because it plays a role in the regulation of ion channels, Ca2 homeostasis, contractility, energetics, cell growth, and it has antioxidant effects and prevents endothelial cells from oxidative stress [6].

The “gold standard” of cardiorespiratory exercise testing—a test with increasing load—allows you to determine the maximum oxygen uptake, assess the level of aerobic capacity of the body and identify the reasons for limiting physical performance [7]. Currently, there are many protocols for exercise testing. However, at different stages of the test, there is practically no information about the functional parameters of the cardiorespiratory system and human biochemical markers, such as heart rate, blood pressure, QRS complex, QT interval, oxygen uptake, carbon dioxide production, respiratory rate and the levels of stable nitric oxide metabolites (nitrites, nitrates) and lactate. This significantly complicates the analysis, comparison and prediction of available data. The purpose of our study was to identify the features of metabolic regulation in highly trained cross-country skiers of different qualifications at different stages of testing at physical maximum load. 

## 2. Materials and Methods

### 2.1. Ethical Approval

The Ethics Committee of the Institute of Physiology of the Russian Academy of Sciences, Syktyvkar approved the experimental design and protocol of the study. The study conformed to the Code of Ethics of the World Medical Association (Declaration of Helsinki). The volunteers were made aware of all the information about the experimental protocol, experimental procedures, and probable risks and inconvenience associated with performing the exercise test on a cycle ergometer “until exhaustion”. After the necessary interpretations, the volunteers gave their written informed agreement to participate in the test. Participants were aware that they were free to leave the study at any time and without consequence. 

### 2.2. Participants

The observation group included 124 highly trained cross-country skiers (male, ages 17–24). The group consisted of two subgroups based on their competition performance: Group I includes 61 nonelite athletes who occupied the last ten places at official competitions, and Group II includes 63 elite athletes who occupied the first ten places at official competitions, who were current members of the national team of the Komi Republic and Russia, had no signs or history of chronic diseases. All participants had more than 5 years of cross-country skiing practice as part of their main training schedule and had extensive experience in endurance events.

### 2.3. Experimental Protocol

The study was performed during the morning after a low-nitrate breakfast. It excluded foods and drinks that are the main sources of nitrates in human food (meat and fish products, vegetables (mainly beets, leafy green vegetables), marinades, (spirits, fruit and mineral drinks). Additionally, the night before the test all the participants consumed a standardized meal (1,674,400–1,757,420 kJcal) consisting of (in units of the percentage of the total energy supplied by the entire meal, En%) 78 En% carbohydrate, 14 En% fat and 8 En% protein. The dietary intake of the participants was assessed using a food frequency questionnaire. Calories consumed at breakfast were not standardize by body weight. The height and body weight of the athletes were measured using a medical weight growth meter (Accunig, SELVAS Healthcare, Daejeon, Korea). At rest (sitting), at the anaerobic threshold (AT) level, during peak load and during the recovery period (5th minute) in each of the athletes examined were determined by the following parameters: systolic blood pressure (SBP), diastolic pressure (DBP), heart rate (HR), Q wave, R wave, and S wave (QRS) complex, QT interval (QT), oxygen uptake (V’O_2_), carbon dioxide production (V’CO_2_), respiratory rate (Rer), and the levels of stable metabolites of NO and lactate in capillary blood samples. Blood pressure was measured by the Korotkov method on the right arm using the Microlife model BRAG-1-30 device (Widnau, Switzerland). An electrocardiogram (ECG) was recorded in 12 leads: standard according to Einthoven (I, II, III), reinforced from the extremities according to Goldberger (aVR, aVL, aVF), and thoracic according to Wilson (V1-6). Manual measurements of ECG characteristics were performed using a ruler for measuring and evaluating electrocardiograms from “Heinrich Mack Nachf” (Karlsruhe, Germany). Heart rate, oxygen uptake, carbon dioxide production, and respiratory rates were obtained from test protocols.

### 2.4. Exercise Test on a Cycle Ergometer “until Exhaustion”

On an ergometer bike (“Ergose-lect-100”, Ergoline GmbH, Hoechberg, Germany) was performed aerobic capacity (V’O_2_ max) testing in the “breath by breath” mode. The parameters were averaged over 15-s segments. The test included one minute of cycling without load (for adaptation of participants) followed by stepwise load increases of 40 W in 2 min increments. The first load started from 120 W. During the test, the pedalling speed was 60 rpm. The anaerobic threshold (AT) was determined by reaching a respiratory coefficient of 1 [8,9].

### 2.5. Determination of NO_x_

The NO levels in the plasma were measured using the Griess reaction, by evaluating the stable metabolites of NO, including nitrites (NO(_2_)(−)) and nitrates (NO(_3_)(−)), which were pre-sorted as an index [NO_x_]. As described earlier in our article [5], nitrite and nitrate are the terminal products of NO in human plasma. It is known that a strong correlation between endogenous NO production and NO_x_ levels exist in plasma [10]. Blood samples with a volume of two ml were collected into tubes with heparin and centrifuged for 20 min at 2500× *g*. The separated plasma was stored at −40 °C until analysis. After deproteinization of plasma samples by precipitation in ethanol and centrifugation, the supernatants were incubated for 30 min at 37 °C with vanadium chloride to convert nitrate to nitrite. Next, the samples were mixed with Griess reagent. Samples were measured at a wavelength of 540 nm using a Spectronic Genesys-6 Spectrophotometer (Thermo Electron Scientific Instruments LLC, Madison, WI, USA). Total nitrite was measured using the Griess reagent. Samples were measured twice against a standard nitrite curve with a known concentration. The plasma nitrate concentration was calculated by subtracting the primary nitrite concentration from the total nitrite concentration. All chemicals used for NO determination were obtained from Sigma (St. Louis, MO, USA). The detection limit for NO was 0.001 µmol/L. The NO_3_/NO_2_ index was calculated as the ratio between NO_3_ and NO_2_.

### 2.6. Determination of Lactate

Plasma lactate levels were measured using a lactate pulmonary alveolar proteinosis (PAP) enzymatic colorimetric method with “Chronolab” commercial kits (Chronolab Systems, S.L. Barcelona, Spain) with an intra-assay coefficient of variance (CV) of 8%. Measurements were performed using a spectrophotometer at a wavelength of 546 nm.

### 2.7. Statistical Analysis

Statistica software (STA862D175437Q, version 6.0, StatSoft Inc., 2001, Tulsa, OK, USA) was used for statistical analysis. The mean (Me) and standard deviation (SD) were calculated. Differences in the dynamics of each parameter were tested by Friedman’s ANOVA. The Wilcoxon test was used to define the correlation coefficients between two variables. The Spearman rank analysis determined the correlation coefficients between two variables. A value of *p* < 0.05 was accepted as statistically significant.

## 3. Results

The characteristics of the examined groups of athletes are presented in Table 1.

The groups of the examined athletes did not have significant differences in age, weight or height (Table 1). At the same time, load power on the anaerobic threshold and maximal load power increased significantly (*p* < 0.05) with an increase in sports activity (Figure 1). 

When performing the test “until exhaustion”, all cross-country skiers showed a statistically significant increase in oxygen uptake during the passage of the anaerobic threshold (*p* = 0.001) compared with indicators at rest (Figure 2). At the level of the AT and peak load, group II was characterized by higher values of oxygen uptake (*p* < 0.01 and *p* < 0.001, respectively). During the recovery period, oxygen uptake values decreased significantly in both groups (*p* < 0.001). During rest and during the recovery period, the oxygen uptake did not differ significantly in the study groups.

### 3.1. Cardiorespiratory Parameters

All cross-country skiers showed some similar dynamics of SBP during the test “until exhaustion”: an increase at AT and at peak load and a decrease at recovery (*p* < 0.001) (Table 2). At rest, Group I was characterized by higher SBP than Group II (*p* < 0.05).

During the period of AT, at the peak load and at the recovery period, the DBP was significantly higher in the athletes of Group II than in the skiers of Group I (*p* < 0.01; *p* < 0.05 and *p* < 0.01, respectively) (Table 2). The level of AT in Group I was decreased in DBP compared with the rest of the period (*p* < 0.01). Athletes in both groups showed an increase in DBP at the peak load compared with the AT Group (*p* < 0.001) and a decrease after five minutes of completion of the test (*p* < 0.001). There were no differences in DBP at rest or during the recovery period between the study groups (*p* > 0.05).

All cross-country skiers showed some similar dynamics of heart rate during the test “until exhaustion”: an increase at the AT and at peak load and a decrease at recovery (*p* < 0.001) (Table 2). At rest and during the recovery period, Group I was characterized by a higher heart rate than Group II (*p* < 0.01 and *p* < 0.05, respectively).

The dynamics of changes in the QRS complex in the two groups were similar; specifically, significant increases during the period of AT passage (*p* < 0.001) and at peak load (*p* < 0.05) and a decrease 5 minutes after the end of the test was noted (*p* < 0.001) (Table 2). There were no significant differences in the QRS complexation during the test “until exhaustion” between the study groups (*p* > 0.05). A significant decrease in QT interval was found in both groups of cross-country skiers during the passage of the AT and at the recovery period (*p* < 0.001) and an increase at the peak of the load (*p* < 0.05). Significant differences in QT interval between groups were found only at rest (*p* < 0.001).

All cross-country skiers showed some similar dynamics of carbon dioxide production and respiratory rate during the test “until exhaustion”: an increase at AT and at peak load and a decrease at recovery (*p* < 0.001). AT Group II was characterized by higher carbon dioxide production than Group I (*p* < 0.01).

### 3.2. Biochemical Parameters

At rest and during the test “until exhaustion”, the values of NO_x_ in Group II were significantly higher than the values of NO_x_ in Group I (*p* < 0.01) (Figure 3). For athletes, Group I and Group II showed an increase in NO_x_ at the AT compared with the rest (*p* < 0.001 and *p* < 0.05, respectively) and a decrease for athletes in group II at peak load (*p* < 0.05). During the recovery period, the NO_x_ level in the examined groups did not change compared to the peak load.

At rest, the NO_2_ value in Group II was significantly higher than the NO_2_ value in Group I (*p* < 0.05) (Table 3). The NO_2_ value between the studied groups of athletes at the AT, at peak load and at the recovery period did not show statistically significant changes (*p* > 0.05). A significant increase in the level of NO_2_ was detected only in Group I during the AT period compared with rest (*p* < 0.01). At rest and during the test “until exhaustion”, the values of NO_3_ in Group II were significantly higher than the values of NO_3_ in Group I (*p* < 0.01). A statistically significant (*p* < 0.01–0.001) increase in NO_3_ was observed in all study groups during the passage of AT compared with the values at rest. In Group II, during the passage of the load peak, the NO_3_ value decreased significantly in comparison with the AT (*p* < 0.01). At the same time, in Group I, a statistically significant decrease in NO_3_ was observed during recovery compared to the peak load (*p* < 0.05).

The subjects of both groups showed no changes in the dynamics of the NO_3_/NO_2_ index during the test “until exhaustion”. At the same time, Group II was characterized by higher values of the NO_3_/NO_2_ index than Group I (*p* < 0.05).

In representatives of different sports qualifications, the level of lactate in the blood increased during the test, while it did not recover after 5 min of the end of the test (*p* < 0.001) (Table 3). At rest in Group I, the lactate value was higher than the lactate value in Group II (*p* < 0.01).

### 3.3. Interrelationship of Cardiorespiratory Parameters and Biochemical Parameters

Correlation analysis between biochemical and cardiorespiratory parameters at different stages of the cross-country skiers in Groups I and II are presented in Table 4 and Table 5, respectively. After correlation analysis, skiers of Groups I and II at rest showed a negative relationship between the values of NO_x_ and lactate (r = −0.26, *p* < 0.05; r = −0.44, *p* < 0.001, respectively), and Group II had a negative relationship between the values of NO_3_ and lactate (r = −0.30, *p* < 0.05) (Table 4). However, in Group II, the relationship between NO_x_ and lactate during the AT period became positive (r = 0.30, *p* < 0.01), and during the peak load period, the relationship between NO_x_ and lactate again became negative (r = −0.26, *p* < 0.05). In Group I, during the period of the peak load, a positive relationship was found between lactate and NO_2_ (r = 0.39, *p* < 0.01), and a negative relationship was found with the NO_3_/NO_2_ index (r = −0.35, *p* < 0.01). During the recovery period, no correlations were found between nitric oxide and lactate in the examined groups. In general, during the AT period, most of the correlations between nitric oxide and cardiorespiratory parameters were found in Group II.

Correlation analysis between maximal oxygen uptake and cardiorespiratory parameters at different stages of the cross-country skiers in Groups I and II are presented in Table 6.

During the AT, the peak load and during the recovery period showed a positive relationship between the maximal oxygen uptake and lactate in Group I (r = 0.26, *p* < 0.05, r = 0.33, *p* < 0.01, r = 0.33, *p* < 0.01, respectively). In Group II, no correlations were found between maximal oxygen uptake and cardiorespiratory parameters.

## 4. Discussion

The main goal of our study was to identify the features of metabolic regulation in highly trained cross-country skiers of different qualifications at different stages of testing at physical maximum load. The study showed that elite athletes with higher results at official competitions were characterized by a higher anaerobic threshold and maximal oxygen uptake. Compared to the group, the athlete was not elite, which is comparable to the literature data [11,12].

Cross-country skiers are known to have a very high maximal oxygen uptake and have equally trained upper and lower body muscles [4]. Thus, they are able to perform submaximal exercise at a rather high metabolic rate and with cardiac output levels similar to or higher than the cardiac output levels achieved by untrained humans at maximal exercise. Thus, this group of athletes is a successful model for studying metabolic effects in humans during intense physical work, especially since trained skiers can actually perform this intensive work, revealing subtle regulatory mechanisms. For example, elite Swedish skiers had a maximum oxygen uptake of 5.1 ± 0.3 L/min [13], which is 7.8% more than among the Group II skiers we examined. Our data on the maximum oxygen uptake also differ significantly in elite athletes from V’O_2_ max of Norwegian representatives of world-class winter sports [14,15], which are characterized by a maximum V’O_2_/kg from 80 to 90 mL/min/kg or 6.5 L/min, which is higher than in Group II surveyed by us by 27.6%. Most likely, this difference can be explained by different methodological approaches for determining the V’O_2_ max and higher anthropometric indicators of elite Swedish skiers (height 180 ± 2 cm, weight 74 ± 2 kg) [13] and Norwegian representatives of world-class winter sports [14,15].

The physical efficiency of athletes and the state of their cardiorespiratory system play leading roles among cross-country skiers in achieving high sports results. Physical aerobic exercise influences vascular remodelling, promoting angiogenesis, positively affecting the number of capillaries and therefore the gas exchange area, while improving oxygen diffusion and increasing vagus tone [14]. In the athletes examined in our study, SBP at rest corresponded to the norm [16]. At the same time, in Group I at rest, the SBP was statistically higher (*p* < 0.05) than in Group II, but during the exercise test, significant differences in SBP between the groups disappeared. A higher SBP in skiers of the 1st group, compared with Group II, could be associated with the incompleteness of the processes of the formation of the cardiovascular system against the background of significant sports loads.

At rest, DBP in all groups of cross-country skiers was above the norm, in comparison with the data obtained from students from a physical education department (64.0 ± 4.7 mm Hg) [16]. It is known that prolonged training in open cold air can lead to an increase in peripheral vascular resistance and, as a consequence, to an increase in DBP [17]. Short-term (one-hour) cold exposure induces hypercoagulation in young healthy people, which can also cause a higher level of DBP [18]. According to the literature [19], cold air can indirectly lead to an increase in cardiovascular risks through its effect on the sympathetic and renin-angiotensin systems, blood pressure, and risk factors for atherosclerosis, such as blood viscosity, the amount of fibrinogen, lipids and uric acid. In our study, during the period of AT, at the peak load and at the recovery period, the DBP was significantly higher in the athletes of Group II than in the skiers of Group I (*p* < 0.01; *p* < 0.05 and *p* < 0.01, respectively) (Table 2).

According to our data, the heart rate at rest in the skiers of Group I was 10.7% higher than the heart rate at rest in Group II (*p* < 0.01), which indicates the formation of bradycardia as a result of sports training, which was more pronounced with an increase in sports activity. This assumption is confirmed by the higher values of the QT interval at rest in Group II compared with Group I. Aerobic exercise [20] affects the parasympathetic nerve, reducing the heart rate, which has a positive effect on reducing cardiovascular diseases. With an increase in sportsmanship, lower heart rate values may indicate [3] higher functional reserves.

On the one side, it was shown that low intensity exercise could improve antioxidant defences and lower lipid peroxidation levels [21]. On the other side, it is known that in the body of professional athletes under intense and strenuous physical exertion, oxidative stress [22] can occur, leading to the accumulation of lipid peroxidation products, including free radicals. Oxidative stress is the main reason for the decrease in the activity of NO synthase (NOS) through a decrease in the availability of the cofactor NOS-tetrahydrobiopterin and, subsequently, the inhibition of the enzymatic synthesis of NO. In our study, lower values of the NO_x_ level during the test “to exhaustion” were observed in Group I skiers compared with athletes in Group II, which may indicate a decrease in the enzymatic synthesis of NO in athletes of Group I. The adaptive capacity of the body reduction with a decrease in the level of NO in the tissues, and pathological changes in metabolism are observed, leading to diseases. The primary cause of the pathogenesis of coronary heart disease and atherosclerotic vascular damage is a deficiency of NO in the vascular endothelium and myocardium [23]. There are several factors causing endothelial NO deficiency: a decrease in eNOS activity [24], destruction or capture of NO by free radicals, and/or a weakening of the effect of NO on smooth muscle [25].

In general, oxidative stress and hypoxia are believed to cause overproduction of NO, often exceeding its physiological level [26,27]. In the body under oxidative stress, NO processes take place; for example, NO covalently binds to a cysteine residue in the beta-chain of Hb to form S-nitrosohaemoglobin, as well as other proteins, which has a regulatory effect on the local tissue blood supply during hypoxic vasodilation [28]. With gradual adaptation to oxidative stress (hypoxia), the plasma level of stable NO metabolites—nitrates and nitrites—progressively increases, correlating with an increase in the vascular NO depot [29]. Perhaps this mechanism of adaptation to the gradually increasing hypoxia caused by physical exertion of maximum power was observed in the elite athletes we examined.

Physiological effects [30] aimed at improving oxygen supply during hypoxia are well documented and include increased ventilation and cardiac output, erythropoiesis, and tissue vascularization [31]. Since V’O_2_ is determined by the interaction of several factors—blood flow, blood O_2_ carrying capacity, diffusion of O_2_ from blood to tissues, ATP demands and O_2_ utilization by mitochondria—it is clear that the result of changes in NO production on V’O_2_ depends on the balance between often opposite effects at different levels of the phenomena occurring.

Lactate levels in blood and tissues are assumed to increase during hypoxia. After 8 h of breathing in a chamber with hypoxic conditions of 12% O_2_, a moderate increase in lactate levels of 29% was observed [32]. In our study, athletes of different sports classifications showed an increase in the level of lactate in the blood during the period of maximum exercise. Compared with the indices at rest in Group I, lactate increased by 3.3 times, and in Group II by 5.2 times (*p* < 0.001); therefore, the athletes examined by us experienced hypoxia during physical activity. Moreover, in Group II, hypoxia had a more pronounced character and a higher increase in V’O_2_ during the period of maximum load. As shown earlier, including our own study, fluctuations in the level of NO in the human body trigger various adaptive reactions under conditions of acute hypoxia [33,34].

Our data indicate that the baseline lactate level before the test “until exhaustion” is also significant, and this indicator is negatively correlated with the NO_x_ indicator. In the literature, in some clinical conditions, there is a negative correlation between lactate and NO; for example, in acute brain injury [35] and in major surgical operations [36], although with septic shock, the use of L-NMMA, an iNOS inhibitor, increases the level of lactate in the thigh muscle [37]. NO is a mediator of skeletal muscle function, especially NO, which affects cellular respiration and contractility, and in working skeletal muscle, inhibition of NOS improves the economy of muscle contraction and leads to a decrease in the outflow of lactate from the muscles by reducing oxygen cost [38].

Thiol reactions or reactive metal centres in proteins can cause NO responses for further biological events in skeletal muscle. The NO-mediated response inhibits haeme-containing proteins such as cytochrome c oxidase, thus inhibiting the function of cytochrome c oxidase and cell respiration [39]. Cytochrome c oxidase and the sarcoplasmic reticulum Ca21-ATPase in fast-twitch and slow-twitch skeletal muscle inhibits by NOS activity which in turn leads to inhibiting mitochondrial respiration in skeletal muscle [40]. Moreover, aconitase and complex I of the respiratory chain can be additional targets of NO [41]. NO is crucial for the activation and inhibition of ryanodine receptors (RyRs) [42]. For muscle contraction and excitation, the release of Ca^2+^ into the cytosol is necessary, the RyR play a decisive role in this process. In recent years, special properties of NO_2_ have been shown, which make it possible to recognize it as the most significant biologically active signalling molecule [43]. The significant vasodilatory response observed in vivo and in vitro experiments upon administration of NO_2_ solutions suggests that it can be an alternative source of NO [44]. NO_2_ was also reported to participate in adaptation to physiological conditions of hypoxia, for example, caused by physical exertion [45]. Modern concepts of nitrite-dependent mechanisms of adaptation to hypoxia are based on data on the participation of NO_2_ in oxygen-dependent and hypoxia-dependent nitrite reductase processes. NO released as a result of these processes is involved in the regulation of vascular tone, modulation of mitochondrial redox reactions [46], changes in the sensitivity of heart contractile proteins to oxygen [47] and calcium ions [48], and inhibition of induced NOS. Due to the cyclic metabolic transformations of NO, NO_2_ and NO_3_, the optimal level of NO is maintained, which is necessary for the normal functioning of the cardiovascular system in conditions of impaired functioning of NO synthases. However, excess NO is removed by the formation of an NO depot in the form of NO_2_, which protects tissues from oxidative and nitrosative stress. The source of vasoactive NO has now been established to also be NO_2_. It is always present in the blood and can be enzymatically reduced to NO under the action of xanthine oxidoreductase and nonenzymatic under conditions of low pH and pO2 [49]. Nitrate is reduced to nitrite and nitric oxide, which activate soluble guanylate cyclase [50]. Exercise has been shown to increase plasma nitrite levels by increasing NO synthesis in endothelial cells [51,52]. Plasma nitrite is oxidized to nitrate. This process is significantly accelerated by the presence of haeme proteins [53]. Nitrates are stable in plasma until excreted in the urine. The circulation of nitrite, rather than nitrate, indicates endothelial-dependent synthesis of NO [54]. Increased NO production during exercise is likely controlled by increased nitrate excretion as a possible mechanism for controlling plasma homeostasis [55]. When oxygen pressure decreases, nitrite reduction by deoxyhaemoglobin produces NO. The generation and release of NO by erythrocytes, along with the oxygen concentration gradient, can be associated with the role of nitrite bound to erythrocytes in the processes of vasodilation in response to hypoxia.

The indicators of correlation between the studied indices are of particular interest for discussion. Notably, there was a positive relationship between the parameters of V’O2max and lactate in Group 1 at all stages of testing. However, in Group II, correlations between these indicators were not revealed at all.

It is also necessary to consider the correlation relationship between indicators of metabolites of nitric oxide and lactate during testing. Initially, at rest, NO_x_ and lactate in both groups showed a significant negative correlation. In Group I, at the AT, correlation disappeared, and at the maximum load, the correlation was revealed between NO_2_ and lactate opposite direction. In Group II, at the AT, there was a positive relationship between NO_x_ and lactate indicators, which inverted at maximum load. Thus, among skiers of Group II, a positive correlation of NO and lactate indicators was detected at the AT, and in Group I, the positive correlation of NO_2_ and lactate indicators was detected only at the maximum load. This observation suggests the existence of an adaptive mechanism for regulating the level of lactate at the AT in highly skilled skiers. The methodology of the training process in cyclic sports is known to be based on the principle of increasing the aerobic performance of the body of athletes. The phenomenon identified by us, in our opinion, may be reflects the positive effect of NO on the production of lactate, which provides an increase in aerobic performance. At the same time, at maximum load, on the contrary, it is required to maximize the activity of glycolysis and, accordingly, the production of lactate. In our opinion, activation of the reroute pyruvate away from pyruvate dehydrogenase (PDH) in an NO-dependent mechanism may be a possible mechanism explaining what is happening at the testing stage of highly skilled skiers at the AT (thereby promoting glutamine-based anaplerosis). The capability of this process was established in a recently published post [56]. To a certain extent, this hypothesis can be evidenced by our materials presented in Table 6. At rest, there was no correlation between lactate and V’O_2_ max in either group. However, in Group I, during exercise, a positive correlation of V’O_2_ max and lactate indices was manifested, which increased during the test until the recovery period. In contrast, in Group II during the entire test, there was no significant correlation between V’O_2_ max and lactate. In our opinion, this lack of significant correlation indicates that the level of lactate in the blood of highly qualified skiers–racers, in contrast to less qualified athletes with submaximal and maximum physical activity, does not depend on the parameters of the V’O_2_ max. It is likely that these athletes have developed additional adaptive mechanisms for regulating the production of lactate, one of which may be the activation of pyruvate abstraction through pyruvate dehydrogenase under conditions of aerobic work.

## 5. Conclusions

The results of our research indicate that the studied groups of athletes with high but different levels of sports qualifications are a convenient model for studying the molecular mechanisms of adaptation to physical loads of maximum intensity. Athletes of higher qualifications reveal additional adaptive mechanisms of metabolic regulation, which is manifested in the independence of serum lactate indicators under conditions of submaximal and maximum power from V’O_2_ max, and they have an NO-dependent mechanism for regulating lactate levels during aerobic exercise, including work at the AT. Limitation: 1. The sample is small to suggest solid conclusions. However, our study included 124 highly trained cross-countries skiers, who, at the time of testing, were current members of the national team of the Komi Republic and Russia. All participants were exposed to hard screening for a number of indicators and were unified. 2. An “until exhaustion” assessment needs further investigation due to the low degree of inter-sample analysis. 3. NO is not the only protagonist to guide the metabolic pathways. In the literature, much attention is paid to studying the role of the participation of nitric oxide interaction during inflammatory in rats [57,58]. At the same time, in our paper, we tried to reveal the mechanisms of the participation of nitric oxide in the process of adaptation to regular, hard and intense physical activity of healthy highly qualified athletes, who have developed specific mechanisms of adaptation to these loads.

## Figures and Tables

**Figure 1 cells-11-00039-f001:**
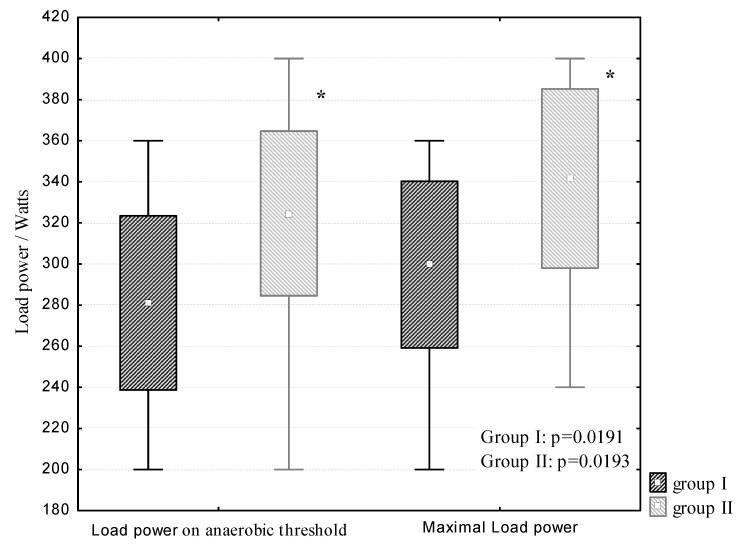
Load power in cross-country skiers during the exercise test on a cycle ergometer “until exhaustion” (Mean; Box: Mean ± 2 SD; Whisker: Min–Max). * *p* < 0.05.

**Figure 2 cells-11-00039-f002:**
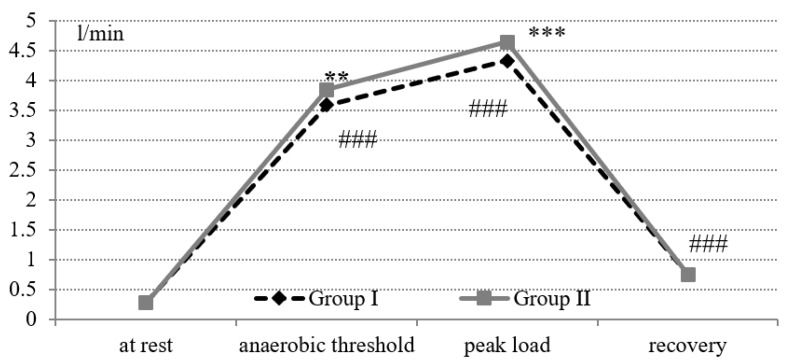
Oxygen uptake during the exercise test on a cycle ergometer “until exhaustion” in cross-country skiers. Statistical significance levels between groups: ** *p* < 0.01; *** *p* < 0.001. Statistical significance levels between stages of the load: ### *p* < 0.001.

**Figure 3 cells-11-00039-f003:**
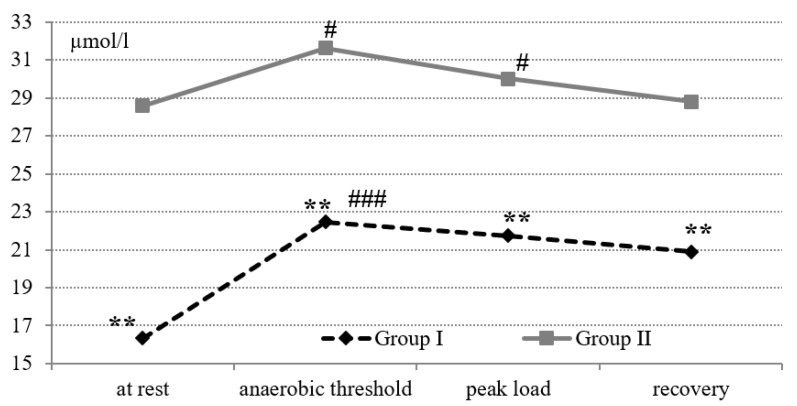
NO_x_ level (µmol/L) during the exercise test on a cycle ergometer “until exhaustion” in cross-country skiers. Statistical significance levels between groups: ** *p* < 0.01. Statistical significance levels between stages of the load: # *p* < 0.05; ### *p* < 0.001.

**Table 1 cells-11-00039-t001:** Characteristics of the subjects by group, ME ± SD.

Parameters	Group I (*n* = 61)	Group II (*n* = 63)
AGE, YEARS	19.1 ± 2.1	21.0 ± 3.1
WEIGHT, KG	69.1 ± 4.8	71.1 ± 4.6
HEIGHT, CM	174.9 ± 4.7	175.4 ± 4.9

**Table 2 cells-11-00039-t002:** Cardiorespiratory parameters at different stages of the load in cross-country skiers, Me ± SD.

Parameters		Stages of the Load			
		At Rest	Anaerobic Threshold	Peak Load	Recovery
Systolic blood pressure, mm Hg	I	118.6 ± 11.7	168.5 ± 16.1 ###	188.6 ± 15.9 ###	122.2 ± 12.2 ###
	Ii	115.2 ± 8.9 *	163.9 ± 13.2 ###	185.8 ± 18.6 ###	125.3 ± 14.3 ###
Diastolic blood pressure, mm Hg	I	77.6 ± 7.6	70.6 ± 13.9 ###	75.2 ± 16.4 ###	62.6 ± 13.5 ###
	Ii	77.9 ± 8.8	77.1 ± 11.5 **	83.4 ± 14.7 *^,^###	67.5 ± 15.4 **^,^###
Heart rate, beats/min	I	62.8 ± 13.1	166.3 ± 13.3 ###	180.2 ± 17.6 ###	108.2 ± 13.3 ###
	Ii	56.1 ± 10.1 **	165.1 ± 15.0 ###	177.8 ± 17.3 ###	99.9 ± 14.6 *^,^###
QRS complex, ms	I	103.8 ± 8.9	191.8 ± 95.8 ###	216.6 ± 76.3 #	123.4 ± 41.9 ###
	Ii	106.4 ± 8.9	187.4 ± 83.6 ###	213.3 ± 87.3 #	113.6 ± 19.7 ###
QT interval, ms	I	394.2 ± 24.9	328.5 ± 93.5 ###	361.8 ± 80.4 #	316.8 ± 41.6 ###
	Ii	412.5 ± 29.4 ***	316.8 ± 84.3.1 ###	345.4 ± 93.6 #	311.1 ± 28.4 ##
Carbon dioxide production, L/min	I	0.3 ± 0.1	3.6 ± 0.5 ###	4.7 ± 0.6 ###	0.8 ± 0.2 ###
	Ii	0.3 ± 0.1	3.9 ± 0.6 **^,^###	4.8 ± 0.7 ###	0.8 ± 0.2 ###
Respiratory rate, breaths per minute	I	13.9 ± 3.8	35.6 ± 8.2 ###	50.2 ± 10.1 ###	25.8 ± 5.8 ###
	Ii	13.6 ± 3.9	36.8 ± 8.7 ###	50.8 ± 12.4 ###	25.7 ± 4.9 ###

Statistical significance levels between groups: * *p* < 0.05; ** *p* < 0.01; *** *p* < 0.001 Statistical significance levels between stages of the load: # *p* < 0.05; ## *p* < 0.01; ### *p* < 0.001.

**Table 3 cells-11-00039-t003:** Nitric oxide and lactate levels at different stages of the load in cross-country skiers, ME ± SD.

Parameters		Stages of the Load			
		At Rest	Anaerobic Threshold	Peak Load	Recovery
NO_2_, µmol/L	I	8.1 ± 3.6	10.2 ± 4.9 ##	9.8 ± 3.9	10.7 ± 4.5
	Ii	11.6 ± 5.2 *	11.6 ± 6.1	11.1 ± 5.4	11.5 ± 5.7
NO_3_, µmol/L	I	8.2 ± 4.0	12.3 ± 6.9 ###	11.9 ± 6.2	10.2 ± 5.8 #
	Ii	17.2 ± 8.6 **	20.0 ± 10.6 **^,^##	18.9 ± 11.3 **^,^#	17.3 ± 10.5 **
NO_3_/NO_2_ index	I	1.5 ± 0.5	1.6 ± 0.9	1.6 ± 0.2	1.3 ± 0.8
	Ii	2.3 ± 0.6 *	2.2 ± 0.6 *	2.2 ± 1.3 *	1.9 ± 0.5 *
Lactate, µmol/L	I	2.9 ± 0.9	6.2 ± 1.6 ###	9.6 ± 2.2 ###	9.7 ± 2.3
	Ii	2.0 ± 0.8 **	6.4 ± 1.8 ###	10.3 ± 1.8 ###	9.8 ± 2.7

Statistical significance levels between groups: * *p* < 0.05; ** *p* < 0.01 Statistical significance levels between stages of the load: # *p* < 0.05; ## *p* < 0.01; ### *p* < 0.001.

**Table 4 cells-11-00039-t004:** Correlations between biochemical and cardiorespiratory parameters at different stages of cross-country skiers in Group I.

Stages of the Load	Parameters	Spearman Rank Order Correlations		
		NO_x_	NO_2_	NO_3_
Before load, at rest	SBP	−0.32 *		
	DBP			
	HR			
	QRS			
	QT			
	Lactate	−0.26 *		
	V’O_2_			
	V’O_2_ max	0.30 *		
	V’CO_2_			
	Rer			
Anaerobic threshold	SBP	−0.25 *	−0.36 **	
	DBP			
	HR			
	QRS			
	QT			
	Lactate			
	V’O_2_	0.36 **	0.28 *	
	V’O_2_ max	0.29 *		
	V’CO_2_	0.34 **		
	Rer			
Peak load	SBP			
	DBP			
	HR			
	QRS			
	QT			
	Lactate		0.39 **	
	V’O_2_			
	V’O_2_ max			
	V’CO_2_			
	Rer			
Recovery	SBP			
	DBP			
	HR			0.29 *
	QRS			
	QT			
	Lactate			
	V’O_2_			
	V’O_2_ max			
	V’CO_2_			
	Rer			

Statistical significance levels: * *p* < 0.05; ** *p* < 0.01. SBP-systolic blood pressure, DBP-diastolic blood pressure, HR-heart rate, QRS-QRS complex, QT-QT interval, V’O_2_-oxygen uptake, V’O_2_ max-maximal oxygen uptake, V’CO_2_-carbon dioxide production, Rer-respiratory rate.

**Table 5 cells-11-00039-t005:** Correlations between biochemical and cardiorespiratory parameters at different stages of cross-country skiers in Group II.

Stages of the Load	Parameters	Spearman Rank Order Correlations		
		NO_x_	NO_2_	NO_3_
Before load, at rest	SBP			
	DBP			
	HR			
	QRS	0.37 **		
	QT			
	Lactate	−0.44 ***		−0.30 *
	V’O_2_			
	V’O_2_ max			
	V’CO_2_			−0.30 **
	Rer	−0.26 *		−0.26 *
Anaerobic threshold	SBP			0.26 *
	DBP			
	HR	0.31 **		0.43 ***
	QRS			
	QT			
	Lactate	0.30 **	0.26 *	
	V’O_2_	0.33 **		0.32 **
	V’O_2_ max			
	V’CO_2_	0.26 *		0.25 *
	Rer	0.26 *		0.26 *
Peak load	SBP	−0.29 *		−0.26 *
	DBP		0.27 *	
	HR			
	QRS			
	QT			
	Lactate	−0.26 *		
	V’O_2_			
	V’O_2_ max	0.26 *		
	V’CO_2_			
	Rer			
Recovery	SBP			
	DBP			
	HR			0.27 *
	QRS			
	QT			
	Lactate			
	V’O_2_			
	V’O_2_ max			
	V’CO_2_			
	Rer	−0.29 *	−0.28 *	

Statistical significance levels: * *p* < 0.05; ** *p* < 0.01; *** *p* < 0.001. SBP-systolic blood pressure, DBP-diastolic blood pressure, HR-heart rate, QRS-QRS complex, QT-QT interval, V’O_2_-oxygen uptake, V’O_2_ max-maximal oxygen uptake, V’CO2-carbon dioxide production, Rer-respiratory rate.

**Table 6 cells-11-00039-t006:** Correlations between maximal oxygen uptake and cardiorespiratory parameters and lactate at different stages of cross-country skiers in Groups I and II.

Stages of the Load	Parameters	Spearman Rank Order Correlations		
		Groups I	Groups II	
Before load, at rest	SBP			
	DBP			
	HR			
	QRS			
	QT			
	Lactate			
	V’CO_2_			
	Rer			
Anaerobic threshold	SBP	−0.34 **		
	DBP			
	HR			
	QRS			
	QT	0.26 *		
	Lactate	0.26 *		
	V’CO_2_			
	Rer			
Peak load	SBP			
	DBP			
	HR			
	QRS			
	QT			
	Lactate	0.33 **		
	V’CO_2_			
	Rer			
Recovery	SBP			
	DBP	−0.26 *		
	HR			
	QRS			
	QT			
	Lactate	0.33 **		
	V’CO_2_			
	Rer			

Statistical significance levels: * *p* < 0.05; ** *p* < 0.01. SBP-systolic blood pressure, DBP-diastolic blood pressure, HR-heart rate, QRS-QRS complex, QT-QT interval, V’CO_2_-carbon dioxide production, Rer-respiratory rate.

## Data Availability

The data that were generated and/or analyzed during the current study are available from the corresponding author upon reasonable request.

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
