# Peer review of "Features of Metabolic Support of Physical Performance in Highly Trained Cross-Country Skiers of Different Qualifications during Physical Activity at Maximum Load"

_cells, 2021, doi:10.3390/cells11010039_

Round 1
Reviewer 1 Report
You shold carefully check your conclusions with regard to the explorative correlation analysis and the low correlations. Even the correlations are significant the interpretation should be given with clear reservations.
Minors: Check the spelling and use of abbreviations and units for such high ranking journal in natural science. Examples:
- VO2, VCO2 may indicate a volume; Use V'O2, V'CO2 or V<dot>
- kcal is not a SI standard unit
- mmHg instead "mm Hg"
- consequent indexing for NO2, NO3
There are some sentences which may confuse when referring to the two groups (e.g. 'sportmenship' in ln. 145). There are further typing errors (e.g. Tab. 1: ME for means, Group II)
Ln. 353/354 '...at maximum power...': Are you refering to the test?
Author Response
We are sincerely grateful to the Reviewer for his consideration of our manuscript. We agree with all comments. All specified mistakes in the text were corrected. We hope that our manuscript has become better. More detailed answer in the attached file.

Reviewer 2 Report
Dear Authors,
Thank you for the opportunity to review the manuscript "Features of metabolic support of physical performance in highly trained cross-country skiers of different qualifications during test of physical at maximum load". The paper is well structured, but it is necessary to smooth out some imperfections in the drafting.
- I recommend rephrasing the title and changing the term "physical test at maximum load"
- Line 30 Missing ref. Please cite and discuss the following paper: https://dx.doi.org/10.3389%2Ffcvm.2019.00148
- References 4,5,6 are somewhat dated references and have little impact on the athlete. References 4,5,6 are somewhat dated references and have little impact on the athlete.
I recommend to cite and discuss the following references: https://doi.org/10.3389/fcvm.2020.582021 and https://doi.org/10.1016/j.bbadis.2020.165913
- Line 59 I suggest calling it "Design"
- Line 69 being the methods section, in the paragraph of the participants it is not appropriate to insert the results of the inclusion, but it is necessary to describe how they have been included with the eligibility.
- Line 142 Start the results with the participants included, the characteristics at the baseline
- Figure 1: Described exclusively non-parametric tests, it is improper to represent them in the figure with bar diagrams, I recommend the representation with boxes and whiskers
- Line 261 Begin the discussion with a paraphrase of the goal and then describe the major findings of the results
- Line 280-283. I suggest a statement and ref. as follows: "It is already known that physical activity could improve antioxidant defenses and lower lipid peroxidation levels." Please cite and discuss the following papers: https://doi.org/10.3390/ijms22115722 and https://doi.org/10.3390/nu12020574
- Line 437: The limitation section is totally missing. I would suggest at least: 1. The sample is small to suggest solid conclusions. 2. An "until exhaustion" assessment needs further investigation due to the low degree of inter-sample analysis. 3. NO is not the only protagonist to guide the metabolic pathways, in fact the cytokine. Please cite and discuss the following papers: https://doi.org/10.1111/1756-185X.14194 and https://doi.org/10.3390/antiox9121284
Author Response

(The authors gave the same response as above.)

Round 2
Reviewer 1 Report
The authors followed the advices of the reviewer. There are no further concerns for publication.